# *Bulinus* snails in the Lake Victoria Basin in Kenya: Systematics and their role as hosts for schistosomes

**Caitlin R. Babbitt****[1]***, **Martina R. Laidemitt[1], Martin W. Mutuku[2], Polycup O. Oraro[2], Sara V. Brant[1], Gerald M. Mkoji[2], Eric S. Loker[1]**

**1** Center for Evolutionary and Theoretical Immunology, Division of Parasites, Museum of Southwestern Biology, Department of Biology, University of New Mexico, Albuquerque, New Mexico, United States of America, **2** Centre for Biotechnology Research and Development, Kenya Medical Research Institute, Nairobi, Kenya

* cbabbitt@unm.edu

**Data Availability Statement:** All sequence files are available from the Genbank database (accession

## Abstract

The planorbid gastropod genus *Bulinus* consists of 38 species that vary in their ability to vector *Schistosoma haematobium* (the causative agent of human urogenital schistosomiasis), other *Schistosoma* species, and non-schistosome trematodes. Relying on sequence-based identifications of bulinids (partial *cox1* and *16S*) and *Schistosoma* (*cox1* and *ITS*), we examined *Bulinus* species in the Lake Victoria Basin in Kenya for naturally acquired infections with *Schistosoma* species. We collected 6,133 bulinids from 11 sites between 2014–2021, 226 (3.7%) of which harbored *Schistosoma* infections. We found 4 *Bulinus* taxa from Lake Victoria (*B. truncatus*, *B. tropicus*, *B. ugandae*, and *B. cf. transversalis*), and an additional 4 from other habitats (*B. globosus*, *B. productus*, *B. forskalii*, and *B. scalaris*). *S. haematobium* infections were found in *B. globosus* and *B. productus* (with infections in the former predominating) whereas *S. bovis* infections were identified in *B. globosus*, *B. productus*, *B. forskalii*, and *B. ugandae*. No nuclear/mitochondrial discordance potentially indicative of *S. haematobium*/*S. bovis* hybridization was detected. We highlight the presence of *Bulinus ugandae* as a distinct lake-dwelling taxon closely related to *B. globosus* yet, unlike all other members of the *B. africanus* species group, is likely not a vector for *S. haematobium*, though it does exhibit susceptibility to *S. bovis*. Other lake-dwelling bulinids also lacked *S. haematobium* infections, supporting the possibility that they all lack compatibility with local *S. haematobium*, thereby preventing widespread transmission of urogenital schistosomiasis in the lake's waters. We support *B. productus* as a distinct species from *B. nasutus*, *B. scalaris* as distinct from *B. forskalii*, and add further evidence for a *B. globosus* species complex with three lineages represented in Kenya alone. This study serves as an essential prelude for investigating why these patterns in compatibility exist and whether the underlying biological mechanisms may be exploited for the purpose of limiting schistosome transmission.

numbers OP235425-OP235450; OP234396-OP234421; OP233086-OP233143; OP244902-OP244960; OP242173-OP242177).

**Funding:** This study was funded by the National Institute of Health (https://www.nih.gov/) grant R37AI101438. The funders had no role in study design, data collection and analysis, decision to publish, or preparation of the manuscript.

**Competing interests:** The authors declare no competing interests.

## Author summary

Human schistosomiasis is a neglected tropical disease caused by members of the trematode genus *Schistosoma*. Every schistosome species is dependent on a particular species, or array of species, of intermediate gastropod host(s) for their transmission. In the Lake Victoria Basin in Kenya, two related schistosome species (*Schistosoma haematobium* and *Schistosoma bovis*) utilize multiple species within the genus *Bulinus* as intermediate hosts. Discerning which bulinid species vector *S. haematobium* or *S. bovis*, or both, and identifying the habitats for each, is critical to understanding local transmission patterns. Closely related bulinids cannot be confidently distinguished using morphological criteria so this study used DNA sequence-based methods to identify local bulinid species and to identify schistosomes shed from infected snails. We implicate two bulinid species in the transmission of *S. haematobium* and four species in the transmission of *S. bovis*. Both *S. haematobium* associated species were found exclusively in streams and dams in the Lake Victoria Basin thereby seemingly keeping the shores of Lake Victoria largely free of *S. haematobium* transmission. Further study as to why some species like *B. globosus* are susceptible to *S. haematobium* whereas other close relatives like *B. ugandae* are apparently refractory may reveal underlying resistance factors potentially useful for control programs.

## Introduction

One of the fascinating aspects of the biology of infectious diseases is that, in some cases, the parasites responsible and their hosts (including vectors) do not comprise a single parasite or host species, but complex arrays of related species [1–4]. Such arrays might reveal a checkerboard of host-parasite interactions, ranging from pairs of host and parasite species being fully compatible and supporting transmission to marginally compatible and fully incompatible pairs. The *Schistosoma haematobium* species group and species within the genus *Bulinus* comprise such an array [5]. We have directed our attention to representatives of these two groups of organisms that occupy the Kenyan waters of the Lake Victoria Basin (LVB), in hopes of eventually revealing the factors that dictate the various outcomes of such associations.

Like most digenetic trematodes, schistosomes depend on a gastropod intermediate host to complete their life cycles, within which vertebrate-infective cercariae are asexually produced in prolific numbers. Such gastropods are often termed intermediate hosts because of their obligatory role in schistosome larval development, and although they do not directly deliver parasites to their vertebrate hosts as, for example, a mosquito transmits malaria parasites, they nonetheless play an indispensable vector role. Successful transmission of vector-borne parasites like schistosomes is dependent on a variety of factors, including ecological circumstances which impact the encounter rates between parasite and vector [6–8], associations with particular symbionts including those that prey on the free-living forms of certain parasites [9], facilitated susceptibility where prior infection with a specific parasite allows a second parasite to develop in a non-typical host [10,11], and nuanced physiological and immunological interactions which dictate the outcome of an infection [2,12–20].

To begin to fully appreciate the intimate relationships between vector and parasite that influence compatibility and ultimately transmission, a sound understanding of the underlying systematics of both parasite and vector host is critical. This task is complicated when the species involved cannot consistently be accurately differentiated using morphology alone, or when a lack of clear morphological differences belie the presence of large genetic differences, that is, when cryptic species are involved [21].

With rare exceptions, members of the *Schistosoma haematobium* group depend on freshwater snails of the planorbid genus *Bulinus* for their transmission. This species group includes 9 species: *S. margrebowiei*, *S. leiperi*, *S. mattheei*, *S. intercalatum*, *S. guineensis*, *S. curassoni*, *S. bovis*, *S. kisumuensis*, and *S. haematobium* [22–24]. Collectively, they pose a persistent threat to human and domestic animal health throughout Africa, in parts of the Mediterranean region, and the Middle East. The agent of urogenital schistosomiasis, *S. haematobium*, is the most common human schistosome [25]. In general, anatomical similarities and a history of apparent hybridization among members of the *S. haematobium* group, [12,26–30] coupled with the changes posed by present-day events such as climate change [31] further highlight the need to clarify both the systematic status of schistosomes and the snails that transmit them.

There are 38 currently recognized species of *Bulinus* [32–35]. Bulinids have proven particularly challenging to identify because variable morphological and conchological traits make species identification and differentiation difficult [32,36–39] and the group is inherently complex with various mating systems represented [40] and some polyploid species [41].

Means to differentiate and reliably identify bulinid snails have improved considerably with the use of sequence-based genetic markers [34,38,42–44], mitochondrial genomes [35,45,46], and a recently published nuclear genome [47]. Modern phylogenetic analyses have provided a more firm systematic foundation for *Bulinus* [34,38,44–46,48,49], improved discrimination among morphologically similar species [39,50], better defined the four species groups within *Bulinus* [34,38], and improved understanding of how the genus diversified and evolved [34,35,44,45,51,52]. Such contributions have provided tools to determine what particular *Bulinus* species are involved in the transmission of the various species of *Schistosoma* [49,53,54] and other trematodes including several species of livestock-infective amphistomes, [55,56], and echinostomes [57]. Such investigations have greatly expanded our knowledge regarding the diversity of parasites that a snail species can transmit and have additionally revealed novel and presently unstudied parasite species [57,58].

As shown in numerous studies to date, the relationships between the *S. haematobium* group species and *Bulinus* species are complex [5,32]. A particular species of *Bulinus* may act as vector for multiple schistosome species, a single schistosome species, or not be involved in schistosome transmission at all [59–61]. In some cases, local adaptation of bulinids and schistosomes has resulted in a given schistosome species utilizing different intermediate host species in different regions. Previously, this was thought to be due to the existence of at least two *S. haematobium* strains which differ in their compatibility with *Bulinus* species [62–64]. More recently, it has been shown that *S. haematobium* isolates across Africa have low genetic diversity as compared to *S. bovis*, and that all tested isolates of *S. haematobium* (with the exception of the Madagascar isolate) have been observed to contain various levels of *S. bovis* introgression in their genomes [28–30]. Variable intermediate host-use patterns among modern *S. haematobium* isolates may be influenced by the particular segments of the *S. bovis* genome they retain [30].

The focus of this study is primarily on the relationships between bulinids and schistosomes and includes some insight into the relationships between bulinids and non-schistosome trematodes, in western Kenya, in the LVB. Lake Victoria is the world's largest tropical lake and connects Kenya, Tanzania, and Uganda, with the surrounding LVB including a variety of smaller water bodies such as streams, dams, papyrus swamplands, rain-fed pools, ponds, and springs [65]. Based on considerations of conchology, anatomy, ploidy, and enzyme electrophoresis, Brown [32] recognized 38 *Bulinus* species divided into 4 species groups. 12 of which he reported from the LVB: from the *B. africanus* group, *B. africanus*, *B. nasutus productus*, *B. globosus* and *B. ugandae*; from the *B. forskalii* group, *B. forskalii*, *B. browni* and *B. scalaris*; from the B. *tropicus/truncatus* group, *B. transversalis*, *B. tropicus*, *B. truncatus* and *B. trigonus;* and

from the *B. reticulatus* group, *B. reticulatus*. Several of the species he reported are hard to differentiate from one another, and some are rarely encountered or studied and in general are poorly known, including *browni*, *scalaris* and *reticulatum*. More recently, based largely on the useful discrimination provided by the cytochrome c oxidase subunit 1 (*cox1*) gene, Chibwana *et al.* [50] found 7 species in the LVB, including *B. globosus* (described as a complex), *B. truncatus*, *B. tropicus*, *B. nasutus productus* and *B. forskalii* as well as two taxa, *Bulinus* sp. 1 and 2, provisionally identified as *B. trigonus* and *B. ugandae*, respectively. Currently accepted species are associated (with some variation) with ephemeral pools or ponds (e.g. *B. forskalii*, *B. scalaris*, *B. reticulatus*), seasonal ponds or springs (*Bulinus productus*), more permanent habits such as streams or dams (*Bulinus globosus*), the lakeshore and associated papyrus swamps (*Bulinus ugandae*), or the deeper waters of the lake (*B. tropicus*, *B. truncatus* and *B. trigonus*). Of particular note is a growing body of evidence that the pan-African species *B. globosus* is not a single species, but a complex of multiple species [35,38,46,50].

Based on an examination of the relevant literature, coupled with sequence data of marker genes to aid in the identification of both snails and the schistosomes they host, we provide an overview of the bulinid species we have recovered from various habitats in the LVB. We highlight some difficulties regarding *Bulinus* systematics and identify some peculiarities regarding the role of bulinids in the transmission of *S. haematobium* and *S. bovis* in western Kenya. This study serves as a prelude to investigations aimed at understanding the underlying causes dictating the patterns of compatibility posed by the complex interacting arrays of *Schistosoma* and *Bulinus* species in western Kenya.

## Materials and methods

### Ethics statement

Informed written consent was obtained from all individual participants included in the study. The Kenya Medical Research Institute Scientific and Ethics Review Unit (KEMRI/SERU/ CBRD/173/3540) and the University of New Mexico Institution Review Board (IRB 821021– 12, IRB 821021–9) approved all aspects of this project involving human subjects. Ethical approval for the collection and analyses of snail and schistosome samples were obtained from the National Commission for Science, Technology and Innovation (permits number NACOSTI/P/21/9648 and NACOSTI/P/22/17142), and National Environmental Management Authority (permit number NEMA/AGR/149/2021).

### Sampling

We collected *Bulinus* snails from 11 different localities (S1 Table and S1 Fig); some localities include endemic transmission sites where we have collected from Jan 2014 –Mar 2021. Two methods were used to collect snails: scooping from the shore and dredging from a boat [66]. From the shore, two experienced lab members scooped snails for 30 minutes per sampling site using long-handled scoops (steel sieve with a mesh size of 2 × 2 mm, supported on an iron frame). Offshore from a boat, snails were collected for 30 min by passing a dredge (0.75 m long and 0.4 m wide with an attached sieve, 2 × 2 mm mesh size) along the bottom. Dredge hauls were made, beginning at 1 m depth and extending perpendicular to the shore to a maximum of 10 m depth, typically covering a distance of about 150 m. Live snails were transported to the Kenya Medical Research Institute (KEMRI), Center for Global Health Research, Kisian, Kisumu.

Snails were provisionally identified using keys [32,67]. Snails were rinsed and placed one snail per well in 12-well cell culture plates in 3 ml of aged tap water. The plates were placed in ambient outdoor lighting for 2 hr to induce cercarial shedding. Cercariae were identified

morphologically [68]. Each shedding snail was preserved in one sample tube, and the cercariae they released in a corresponding tube, all in 95% ethanol. Non-shedding snails were maintained in the lab to allow cercariae-shedding infections to develop and re-shed 1–5 weeks later. Snails were maintained in 20 L tanks with oyster shells, aeration, and fed boiled lettuce and shrimp pellets.

*S. haematobium* miracidia were sourced from the urines of local schoolchildren enrolled in this study (see ethics statement below) or from discarded clinical samples and were used for phylogenetic analyses and comparisons with schistosomes shed from infected snails.

### Additional sampling records

Additional specimens were obtained by a loan from collections held at the Division of Parasites, Museum of Southwestern Biology, University of New Mexico.

### Molecular characterization

**Snail sequences.** Prior to extraction, snails to be processed for sequencing were photographed to provide a record of shell size and shape. Snail genomic DNA was extracted from a small portion of the head foot using the E.Z.N.A. Mollusc DNA Kit (Omega Bio-Tek, Norcross, GA) according to manufacturer's instructions. Partial sequences of the cytochrome c oxidase subunit I (*cox1*) and *16S* rRNA genes were obtained for molecular identification and differentiation among *Bulinus* species.

*Cox1* partial sequences (706 bp) were amplified using universal primers [69] and occasionally using reverse primer COR722b [70]. *16S* partial sequences (481 bp) were amplified using forward primer 16Sar and reverse primer 16Sbr [71]. Thermocycling conditions for both *cox1* and *16S* were as follows: preheat at 94°C for 5 min followed by 45 cycles of denaturation at 94°C for 15 sec, annealing at 45°C for 30 sec and extension at 72°C for 1 min; final extension step at 72°C for 10 min. All snail and parasite PCR reactions had a volume of 25 $\mu$L with 1 $\mu$L of 40 ng of DNA, 0.8 mM/l dNTPs, 2.5 mM/l MgCl$_2$, 0.25 units of Ex Taq DNA (Clontech, Mountain View, CA), and 0.4 $\mu$M/L of each primer.

**Schistosome sequences.** Partial *cox1* mtDNA and partial internal transcribed spacer 1 (ITS1) + 5.8S + partial internal transcribed spacer 2 (ITS2) rRNA sequences were used to identify and differentiate among *Schistosoma* species. A single cercaria was removed from the ethanol preserved cercariae obtained from a single snail and used for DNA extraction. Genomic DNA was extracted from parasite specimens using QIAamp DNA Micro kit (Qiagen, Valencia, CA) according to manufacturer's instructions with a 40 $\mu$L final elution volume.

*Cox1* partial mtDNA (423 bp) sequences were generated using a modified forward primer designed from the *S. bovis/S. haematobium* universal primer [72] (ModShAsmit1: 5' TTTTTTGGKCATCCTGAGGTGTAT3'), and the reverse primer *Cox1*_schist_3' [73]. Thermocycling conditions were as follows: preheat at 94°C for 5 min followed by 30 cycles of denaturation at 94°C for 30 sec, annealing at 40°C for 30 sec and extension at 72°C for 2 min followed by a final extension period of 72°C for 5 min.

ITS1 + 5.8S + ITS2 partial rRNA (981 bp) sequences were amplified using forward primer ITS5 and reverse primer ITS4 [74]. Thermocycling conditions were as follows: preheat at 94°C for 5 min followed by 30 cycles of denaturation at 94°C for 30 sec, annealing at 54°C for 45 sec and extension at 72°C for 1 min; followed by a final extension period of 72°C for 5 min.

**For both snails and schistosomes.** PCR fragments were separated by 1% agarose gel electrophoresis and visualized with 0.5% GelRed Nucleic acid gel stain (Biotium, Hayward, CA). PCR products were purified using ExoSap-IT (Affymetrix, Santa Clara, CA). Both strands were sequenced using an Applied Biosystems 3130 automated sequencer and BigDye

terminator cycle sequencing kit Version 3.1 (Applied Biosystems, Foster City, CA). DNA sequences were verified by aligning reads from the 5′ and 3′ directions using Sequencher 5.1 and manually corrected for ambiguous base calls (Gene Codes, Ann Arbor, MI).

Additional *Bulinus* (MT707420.1, AM286295.2, AM286296.2, LT671915.1, LT671916.1, MK414452.1, MK414453.1, MK414454.1, AM286286.2, AM286299.2, AM286300.2, AM286303.2, AM921814.1, AM286308.2, AM286309.2, MN551559.1, AM286318.2, MT707391.1, MT707392.1, MT707382.1, AM286311.2, AM921838.1, MT707425.1, GU451744.1, MH037061.1) sequences were retrieved from NCBI [38,46,48,50,75–77]. Additional *Schistosoma* sequences were used to represent *S. mattheei* (MW046871.1, AJ519518.1), *S. guineensis* (Z21717.1, AJ519523.1), and S. *curassoni* (MT580946.1, MT579422.1) for the *cox1 + ITS* concatenated phylogenetic analysis [78–81]. *Indoplanorbis exustus* and *Schistosoma mattheei* were selected as outgroups for the *Bulinus* and *Schistosoma* analyses, respectively. Genbank sequences MH037061 and MH037083, and GU451744 and GU451726 were concatenated to produce outgroup sequences for the bulinid *cox1 + 16S* concatenated alignment [76,77]. GenBank accession numbers for bulinid sequences provided in this study can be found in Table 1.

Multiple sequence alignments were performed using the program MUSCLE [82] in MEGA X [83]. The best fit maximum likelihood (ML) nucleotide substitution model was chosen for all genes in MEGA X using BIC criterion. Phylogenetic relationships were inferred using ML in MEGA X using 1000 bootstrap replicates. Uncorrected pairwise distance values (*p*-distances) were calculated in MEGA X [83]. Data were summarized within and between groups (Tables 2 and S2).

Specimens sequenced as part of this study were deposited as vouchers in the Division of Parasites, Museum of Southwestern Biology at the University of New Mexico. Snail and parasites specimens were designated a MSB:Host: or a MSB:Para: number, respectively (Tables 1 and 3).

## Results

### Overview of *Bulinus* collections

A total of 6,133, *Bulinus* snails were collected from 11 locations (S1 Fig) in in the LVB between January 2014 and March 2021 and initially provisionally identified, in some cases just to species group (S1 Table). Recovered snails included *B. globosus* (n = 2994), *B. ugandae* (n = 889), *B. productus* (n = 1302), *B. tropicus/truncatus* group species (n = 245), and *B. forskalii* group species (n = 685). Bulinid species presence and trematode composition and prevalence varied by site (S1 Table). The highest schistosome prevalence was recovered from *B. globosus* at Asao stream (6.5% prevalence) and few to no schistosome infections were recovered from the various lake shore habitats. Further sequence-based specifications of species identities for both bulinids and schistosomes are found below.

### Molecular identification of bulinids

Partial portions of the *cox1* gene were sequenced from 62 bulinids. Because some clades were initially overrepresented, 58 sequences were used in the final phylogenetic analysis (Fig 1). *16S* sequences were produced for 70 bulinid specimens. Some specimens did not produce amplicons for both genes and therefore concatenated (*cox1 +16S*) sequences were produced for 57 bulinid specimens (Fig 2). Specimens were chosen for sequencing to include representative species from the widest variety of habitats possible. Specimen information can be found in Table 1.

**Table 1.** *Bulinus* specimens.

| Species | MSB: Host: | Collection location | Habitat type | Latitude | Longitude | Date (YYYY/MM/DD) | Infection | Cercariae type | GenBank accession COI | GenBank accession 16S |
|---|---|---|---|---|---|---|---|---|---|---|
| *B. globosus* | 24516 | Asao Stream | R | −0.31810 | 35.0069 | 2016-08-02 | y | *Schistosoma* | OP233119 | |
| *B. globosus* | 24525 | Asao Stream | R | −0.31810 | 35.0069 | 2016-08-02 | y | *Schistosoma* | OP233113 | OP244943 |
| *B. globosus* | 24526 | Asao Stream | R | −0.31810 | 35.0069 | 2016-08-02 | y | *Schistosoma* | OP233114 | OP244944 |
| *B. globosus* | 24776 | Asao Stream | R | −0.31810 | 35.0069 | 2015-11-13 | n | | OP233135 | OP244911 |
| *B. globosus* | 24777 | Asao Stream | R | −0.31810 | 35.0069 | 2017-05-20 | y | *Schistosoma* | OP233098 | |
| *B. globosus* | 24778 | Asao Stream | R | −0.31810 | 35.0069 | 2017-05-20 | y | *Schistosoma* | OP233099 | |
| *B. ugandae* | 24542 | Power House | LS | −0.09410 | 34.7076 | 2017-05-23 | y | xiphidiocercariae | OP233117 | OP244923 |
| *B. ugandae* | 24543 | Usenge Beach | LS | -0.072636 | 34.059956 | 2016-04-16 | n | | OP233118 | OP244920 |
| *B. ugandae* | 24544 | Kagwa Beach | LS | −0.356594 | 34.68358 | 2016-04-04 | n | | OP233120 | OP244921 |
| *B. ugandae** | 24558 | Mnazi Mmoja Beach, Ukerewe Island, TZ | LS | -2.1075 | 33.08361 | 2001-04-21 | n | | OP233103 | OP244922 |
| *B. ugandae** | 24559 | Ukerewe Island, Kagera Stream, TZ | R | | | 2001-04-21 | y | strigeid | OP233107 | OP244915 |
| *B. ugandae* | 24549 | Usenge Beach | LS | -0.072636 | 34.059956 | 2017-08-03 | y | echinostome | OP233123 | OP244908 |
| *B. ugandae** | 24573 | ADC Farm, Kisumu | S | -0.3333333 | 34.65 | 1987-01-21 | n | | OP233143 | OP244933 |
| *B. ugandae* | 24550 | Usenge Beach | LS | -0.072636 | 34.059956 | 2016-04-16 | n | | OP233124 | OP244907 |
| *B. ugandae* | 24510 | Koriang Beach | LS | -0.3548806 | 34.65903 | 2017-02-24 | y | xiphidiocercariae | OP233125 | OP244906 |
| *B. ugandae* | 24527 | Dunga Beach | LS | -0.14532 | 34.73633 | 2017-02-07 | y | strigeid | OP242173 | OP244902 |
| *B. ugandae* | 24529 | Kagwa Beach | LS | −0.356594 | 34.68358 | 2016-04-13 | n | | OP233115 | OP244925 |
| *B. ugandae* | 24770 | Gudwa Beach | LS | -0.3573667 | 34.3301 | 2018-11-20 | n | | OP233142 | OP244912 |
| *B. ugandae** | 24564 | Kisumu | LS | −0.1091 | 34.775 | 1987-01-20 | n | | OP242176 | OP244924 |
| *B. ugandae** | 24565 | Kagwell | LS | −0.191111 | 34.503333 | 2005-09-12 | n | | OP233091 | OP244918 |
| *B. ugandae** | 24566 | Nawa | LS | −0.094051 | 34.707601 | 2005-09-28 | n | | OP242177 | OP244919 |
| *B. ugandae** | 24567 | Asembo Bay | LS | −0.1885080 | 34.387534 | 2005-01-20 | n | | OP233092 | OP244932 |
| *B. nasutus** | 24579 | Komarock | | -1.26754 | 36.9094 | 1997-11-12 | n | | OP233139 | OP244916 |
| *B. nasutus** | 24580 | Kyenze | | | | 1997-11-14 | n | | OP233138 | OP244937 |
| *B. nasutus** | 24581 | Ng'alalia | | −1.5357 | 37.2361 | 1997-11-12 | n | | OP233137 | OP244938 |
| *B. productus* | 24561 | Tiengre | EP | -0.0898333 | 34.70313 | 2018-05-25 | n | | OP233131 | OP244950 |
| *B. productus* | 24562 | Tiengre | EP | -0.0898333 | 34.70313 | 2018-05-25 | n | | OP233132 | OP244903 |
| *B. productus* | 24563 | Tiengre | EP | -0.0898333 | 34.70313 | 2018-05-25 | n | | OP233133 | OP244939 |
| *B. productus* | 24522 | Tiengre | EP | -0.0898333 | 34.70313 | 2018-05-22 | y | *Schistosoma* | OP233106 | OP244910 |
| *B. productus* | 24524 | Tiengre, Kenya | EP | -0.0898333 | 34.70313 | 2018-05-21 | y | *Schistosoma* | OP233109 | OP244909 |
| *B. productus* | 24774 | Tiengre | EP | -0.0898333 | 34.70313 | 2018-05-21 | y | bent-bodied strigeid | OP233111 | OP244953 |
| *B. tropicus* | 24551 | Mwea | R | -0.6333333 | 37.46667 | 2015-06-20 | y | echinostome | OP242174 | OP244954 |
| *B. tropicus** | 24766 | Usare | LS | −0.105712 | 34.67429 | 2005-09-08 | n | | OP233094 | OP244929 |
| *B. tropicus** | 24767 | Iringa, Kilima Pond, TZ | P | -7.956333 | 35.86383 | 2001-04-30 | n | | OP233110 | OP244905 |
| *B. tropicus** | 24568 | Sand Harvest, Adupe | L | −0.1013889 | 34.714722 | 2005-09-22 | n | | OP233093 | OP244934 |
| *B. tropicus* | 24545 | Minya Kochillo | L | -0.2363778 | 34.24605 | 2016-09-20 | n | | OP242175 | OP244945 |
| *B. tropicus* | 24546 | Gudwa Beach Dredge | L | -0.3573667 | 34.3301 | 2016-09-20 | n | | OP233121 | OP244946 |
| *B. tropicus* | 24547 | Kadidi Beach | LS | -0.2033583 | 34.15326 | 2016-04-07 | n | | OP233122 | OP244947 |
| *B. tropicus* | 24551 | Mwea | R | -0.6333333 | 37.46667 | 2015-06-20 | y | echinostome | OP233128 | |
| *B. tropicus** | 24552 | Ukerewe Island, Kaseni-Shuleni, TZ | LS | -1.933333 | 32.85 | 2001-04-20 | n | | OP233100 | OP244936 |

(*Continued*)

**Table 1.** (Continued)

| Species | MSB: Host: | Collection location | Habitat type | Latitude | Longitude | Date (YYYY/MM/DD) | Infection | Cercariae type | GenBank accession COI | GenBank accession 16S |
|---|---|---|---|---|---|---|---|---|---|---|
| *B. tropicus** | 24578 | Nyamlebi-Ngoma, Ukerewe Island, TZ | LS | -2.130333 | 3.1685 | 2001-04-22 | n | | OP233105 | OP244904 |
| *B. tropicus** | 24530 | Tala | | -1.270768 | 37.319472 | 1997-11-13 | n | | OP233136 | OP244917 |
| *B. tropicus* | 24574 | Kanyibok | LS | -0.0895806 | 34.08593 | 2017-11-01 | y | echinostome | OP233126 | OP244948 |
| *B. tropicus* | 24775 | Eldoret | D | 0.4671 | 35.3517 | 2014-01-07 | n | | OP233134 | OP244913 |
| *B. tropicus** | 24571 | Mbita Beach | LS | −0.4213889 | 34.2075 | 2005-10-04 | n | | OP233088 | OP244959 |
| *B. truncatus** | 24553 | Ukerewe Island, Kaseni-Shuleni, TZ | LS | -1.933333 | 32.85 | 2001-04-20 | n | | OP233101 | OP244940 |
| *B. truncatus** | 24554 | Kom Ombo, Southern Egypt | LS | 30.54558 | 32.21017 | 2003-03-01 | n | | OP233112 | OP244935 |
| *B. truncatus** | 24569 | Mbita Beach | LS | −0.4213889 | 34.2075 | 2005-10-04 | n | | OP233086 | OP244952 |
| *B. truncatus** | 24570 | Mbita Beach | LS | −0.4213889 | 34.2075 | 2005-10-04 | n | | OP233087 | OP244960 |
| *B. truncatus** | 24572 | Mbita Beach | LS | −0.4213889 | 34.2075 | 2005-10-04 | n | | OP233089 | OP244958 |
| *B. truncatus** | 24517 | Mbita Beach | LS | −0.4213889 | 34.2075 | 2005-10-04 | n | | OP233090 | OP244957 |
| *Bul. cf. transversalis** | 24768 | Usare | LS | −0.105712 | 34.67429 | 2005-09-08 | n | | OP233097 | OP244926 |
| *Bul. cf. transversalis** | 24769 | Usare | LS | −0.105712 | 34.67429 | 2005-09-08 | n | | OP233096 | OP244927 |
| *Bul. cf. transversalis** | 24765 | Usare | LS | −0.105712 | 34.67429 | 2005-09-08 | n | | OP233095 | OP244928 |
| *B. scalaris** | 24555 | Ukerewe Island, Kaseni-Shuleni, TZ | LS | -1.933333 | 32.85 | 2001-04-20 | n | | OP233102 | OP244942 |
| *B. scalaris* | 24514 | Tiengre | EP | -0.0898333 | 34.70313 | 2018-05-25 | y | amphistome | OP233127 | OP244955 |
| *B. forskalii** | 24556 | Ukerewe Island, Mnazi Mmoja, TZ | LS | -2.1075 | 33.08361 | 2001-04-21 | n | | OP233104 | OP244931 |
| *B. forskalii** | 24557 | Safisha Stream, Tunduma, TZ | R | -9.316667 | 32.76667 | 2001-04-29 | y | amphistome | OP233108 | OP244930 |
| *B. forskalii* | 24560 | Tiengre | EP | -0.0898333 | 34.70313 | 2018-05-25 | y | amphistome | OP233130 | OP244949 |
| *B. forskalii* | 24575 | Tiengre | EP | -0.0898333 | 34.70313 | 2018-05-29 | y | pigmented amphistome | OP233140 | OP244951 |
| *B. forskalii* | 24576 | Tiengre | EP | -0.0898333 | 34.70313 | 2018-05-29 | y | pigmented amphistome | OP233141 | OP244941 |
| *B. forskalii* | 24577 | Nyabera | S | −0.1091 | 34.775 | 2018-05-21 | Y | xiphidiocercariae | OP233116 | OP244956 |
| *B. forskalii* | 24771 | Nawa | LS | −0.094051 | 34.707601 | 2016-06-12 | Y | amphistome | OP233129 | OP244914 |

Table 1. Sequenced specimens with associated MSB:Host: numbers, collection locations, habitat type, GPS coordinates (when available), infection status, and associated GenBank accession numbers. Specimen names denoted with * indicate samples from archived specimens. *Bulinus productus* specimens are often designated as *Bulinus nasutus productus* in literature regarding this region. Habitat type abbreviations: LS = lakeshore, L = lake, R = river, EP = ephemeral pond, P = pond, D = Dam, S = Swamp. Samples were collected in Kenya unless otherwise indicated.

## Phylogenetic analysis of bulinids using maximum likelihood methods

From our 11 study sites, we recovered seven named species of *Bulinus* based on *cox1* and *16S* sequences (*B. globosus*, *B. ugandae*, *B. productus*, *B. forskalii*, *B. scalaris*, *B. truncatus*, *B. tropicus*) and one distinct taxon we refer to as *Bulinus cf. transversalis* because it conforms in habitat and conchologically to *B. transversalis* [32] but for which no sequence references currently exist. The sequences we obtained for this taxon did not align with any known bulinid species in GenBank.

**Table 2. Intra- and Interspecies *p*-distance values of partial *cox1* of 68 bulinid sequences.**

| | *B. globosus* | *B. ugandae* | *B. productus* | *B. nasutus* | *B. truncatus* | *B. tropicus* | *B. cf. transversalis* | *B. forskalii* | *B. scalaris* | *B. cf. trigonus* |
|---|---|---|---|---|---|---|---|---|---|---|
| *B. globosus* | **0.0227** | | | | | | | | | |
| *B. ugandae* | 0.0578 | **0.0077** | | | | | | | | |
| *B. productus* | 0.1428 | 0.1178 | **0.0021** | | | | | | | |
| *B. nasutus* | 0.1304 | 0.1083 | 0.0735 | **0.0053** | | | | | | |
| *B. truncatus* | 0.1650 | 0.1397 | 0.1780 | 0.1673 | **0.0095** | | | | | |
| *B. tropicus* | 0.1597 | 0.1375 | 0.1750 | 0.1657 | 0.0342 | **0.0097** | | | | |
| *B. cf. transversalis* | 0.1833 | 0.1576 | 0.1910 | 0.1812 | 0.0485 | 0.0521 | **0.0011** | | | |
| *B. forskalii* | 0.1383 | 0.1294 | 0.1674 | 0.1548 | 0.1778 | 0.1663 | 0.1850 | **0.0039** | | |
| *B. scalaris* | 0.1608 | 0.1408 | 0.1948 | 0.1626 | 0.1762 | 0.1803 | 0.1781 | 0.1126 | **0.0326** | |
| *B. cf. trigonus* | 0.1654 | 0.1424 | 0.1910 | 0.1750 | 0.0486 | 0.0446 | 0.0531 | 0.1718 | 0.1802 | **0.0000** |

Bolded values are intraspecific *p*-distance values. *B. cf. trigonus* consists of GenBank sequences MT707391.1 and MT707392.1 [50].

### *cox1* phylogenetic analysis

A total of 62 *cox1* sequences were generated as a part of this study. Because some clades were overrepresented, 58 *cox1* sequences from this study and 25 sequences from GenBank were used to hypothesize phylogenetic relationships among bulinid specimens we collected. The *cox1* sequence analysis discriminated each of the four *Bulinus* species groups as well as each of the 17 species included in the phylogenetic analysis (Fig 1).

Intraspecific *p*-distances were less than 1% with the exceptions of the East African *B. globosus* complex (2.27%) and *B. scalaris* (3.26%). Interspecific *p*-distances ranged from 5.78% for within species group (ex. *B. globosus* and *B. ugandae*) to 19.48% between species groups (*B. forskalii* and *B. productus*). Exceptionally, members of the *B. truncatus/tropicus* group exhibited low interspecific *p*-distances as compared to other closely related bulinids (Table 2).

### Combined (*cox1* + *16S*) dataset analysis

Concatenated *cox1* and *16S* sequences from GenBank representing outgroup sequences, and 58 sequences from this study (9 species) were used to infer phylogenetic relationships among bulinids. The concatenated sequence analysis discriminated among the three *Bulinus* species groups included in this analysis and additionally allowed interspecific discrimination (Fig 2) with greater resolution than the single *cox1* dataset.

Intraspecific species *p*-distances were less than 1%, with the exception of *B. scalaris*. Interspecific *p*-distances were greater than 5%, with the exception of members of the *B. truncatus/tropicus* group which exhibited low intraspecific *p*-distances (S2 Table).

### Trematode infections in field-collected snails

Natural infection prevalence varied by collection site and by host species (S1 Table). The highest prevalence of patent mammalian schistosome infections was found in *B. globosus* (6.37%) followed by *B. productus* (1.97%) and *B. forskalii/scalaris* (0.71%) with few infections found in *B. ugandae* (0.22%) and no schistosome infections observed among *B. truncatus/tropicus* specimens (S1 Table and Fig 3).

Higher overall trematode diversity was observed among *B. globosus*, *B. ugandae*, and *B. forskalii* (minimum 5 trematode taxa per species) than was observed for *B. truncatus/tropicus* specimens (2 taxa) (Fig 3). This study did not seek to identify non-schistosome trematode cercariae to the species level and has therefore likely underestimated the diversity of trematode

**Table 3. Schistosome samples.**

| Species | MSB: Para: | Collection location | Stage (miracidium, cercaria, adult) | Latitude | Longitude | Date | Host species | GenBank ID COI | Genbank ID ITS |
|---|---|---|---|---|---|---|---|---|---|
| *S. bovis* | 32675 | Tiengre | cercaria | -0.089833 | 34.70313 | 5/21/2018 | *B. productus* | OP235447 | OP234419 |
| *S. haematobium* | 32678 | Asao | cercaria | −0.31810 | 35.0069 | 5/20/2017 | *B. globosus* | OP235442 | OP234418 |
| *S. bovis* | 32679 | Asao | cercaria | −0.31810 | 35.0069 | 5/20/2017 | *B. globosus* | OP235445 | OP234417 |
| *S. haematobium* | 32680 | Asao | cercaria | −0.31810 | 35.0069 | 8/2/2016 | *B. globosus* | OP235444 | OP234408 |
| *S. haematobium* | 32681 | Asao | cercaria | −0.31810 | 35.0069 | 5/20/2017 | *B. globosus* | OP235446 | OP234407 |
| *S. haematobium* | 32682 | Asao | cercaria | −0.31810 | 35.0069 | 8/2/2016 | *B. globosus* | OP235448 | OP234416 |
| *S. haematobium* | 32683 | Asao | cercaria | −0.31810 | 35.0069 | 3/1/2017 | *B. globosus* | OP235441 | OP234415 |
| *S. haematobium* | 32685 | Nyakango School | miracidium | -0.440004 | 34.640005 | 2/14/2019 | *Homo sapiens* | OP235443 | OP234405 |
| *S. haematobium* | 32686 | Asao | miracidium | -0.3169444 | 35.00611 | 11/2/2019 | *Homo sapiens* | OP235440 | OP234401 |
| *S. bovis* | 32687 | Asao | cercaria | −0.31810 | 35.0069 | 11/19/2018 | *B. globosus* | OP235431 | OP234400 |
| *S. bovis* | 32689 | Asao | cercaria | −0.31810 | 35.0069 | 3/19/2019 | *B. globosus* | OP235435 | OP234410 |
| *S. bovis* | 32690 | Asao | cercaria | −0.31810 | 35.0069 | 4/29/2019 | *B. globosus* | OP235430 | OP234413 |
| *S. bovis* | 32692 | Tiengre | cercaria | -0.089833 | 34.70313 | 5/21/2018 | *B. forskalii* | OP235449 | OP234420 |
| *S. bovis* | 32705 | Gudwa Beach | cercaria | -0.3573667 | 34.3301 | 5/15/2019 | *B. ugandae* | OP235434 | OP234409 |
| *S. bovis* | 32693 | Asao | cercaria | −0.31810 | 35.0069 | 3/20/2019 | *B. globosus* | OP235433 | OP234411 |
| *S. bovis* | 32694 | Tiengre | cercaria | -0.08983333 | 34.70313 | 5/18/2019 | *B. forskalii* | OP235436 | OP234412 |
| *S. haematobium* | 32695 | Asao | cercaria | −0.31810 | 35.0069 | 8/2/2016 | *B. globosus* | OP235439 | OP234414 |
| *S. haematobium* | 32696 | Asao | cercaria | −0.31810 | 35.0069 | 3/23/2017 | *B. globosus* | OP235432 | OP234406 |
| *S. haematobium* | 32697 | Nyakango School | miracidium | -0.440004 | 34.640005 | 2/14/2019 | *Homo sapiens* | OP235438 | OP234404 |
| *S. haematobium* | 32698 | Nyakango School | miracidium | -0.440004 | 34.640005 | 2/14/2019 | *Homo sapiens* | OP235437 | OP234403 |
| *S. haematobium* | 32699 | Asao | cercaria | −0.31810 | 35.0069 | 5/20/2017 | *B. globosus* | OP235429 | OP234402 |
| *S. bovis* | 32704 | Asao | miracidium | −0.31810 | 35.0069 | 8/2/2016 | *Bos indicus* | OP235425 | OP234397 |
| *S. bovis* | 32702 | Ayuka Dam | cercaria | -0.449 | 34.65532 | 5/24/2018 | *B. productus* | OP235427 | OP234396 |
| *S. bovis* | 32703 | Tiengre | cercaria | -0.08983333 | 34.70313 | 11/1/2019 | *B. productus* | OP235426 | OP234399 |
| *S. bovis* | 32701 | Tiengre | cercaria | -0.08983333 | 34.70313 | 11/1/2019 | *B. productus* | OP235428 | OP234398 |
| *S. haematobium* | 32706 | Ayuka Dam | cercaria | -0.449 | 34.65532 | 5/24/2018 | *B. productus* | OP235450 | OP234421 |

Sequenced schistosomes with associated MSB:Para: numbers, collection locations, life cycle stage, GPS coordinates (when available), collection date, host species, and GenBank accession numbers.

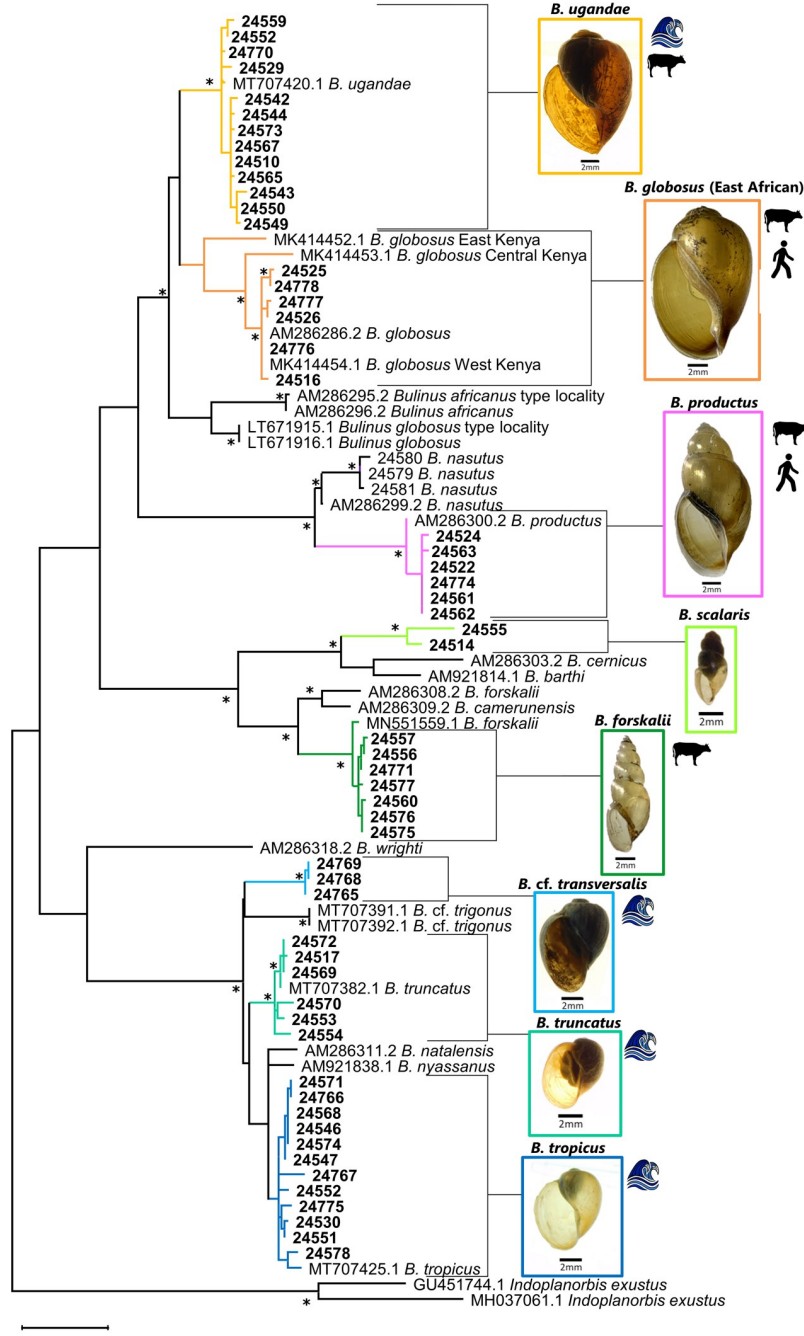

**Fig 1. Phylogenetic relationships among Kenyan bulinids based on partial *cox1* sequences.** Phylogenetic relationships of bulinids from this study and from GenBank (with accession numbers) based on 621 bp of the cytochrome oxidase subunit 1 gene inferred from ML analysis under the GTR+G+I model. Bootstrap values over 95% are indicated by an asterisk. Bolded sequences were generated during this study and listed by MSB:Host: number. Additional information for specimens can be found in Table 1. Specimens recovered within the LVB by this study are color coded by species. *B. africanus* group species are in warm colors, *B. forskalii* group species in greens, and *B. truncatus/tropicus* group species in blues. Wave icons indicates species found within Lake Victoria. Cow icons indicate species with naturally occurring *S. bovis* infections. Human icons indicate species with naturally occurring *S. haematobium* infections.

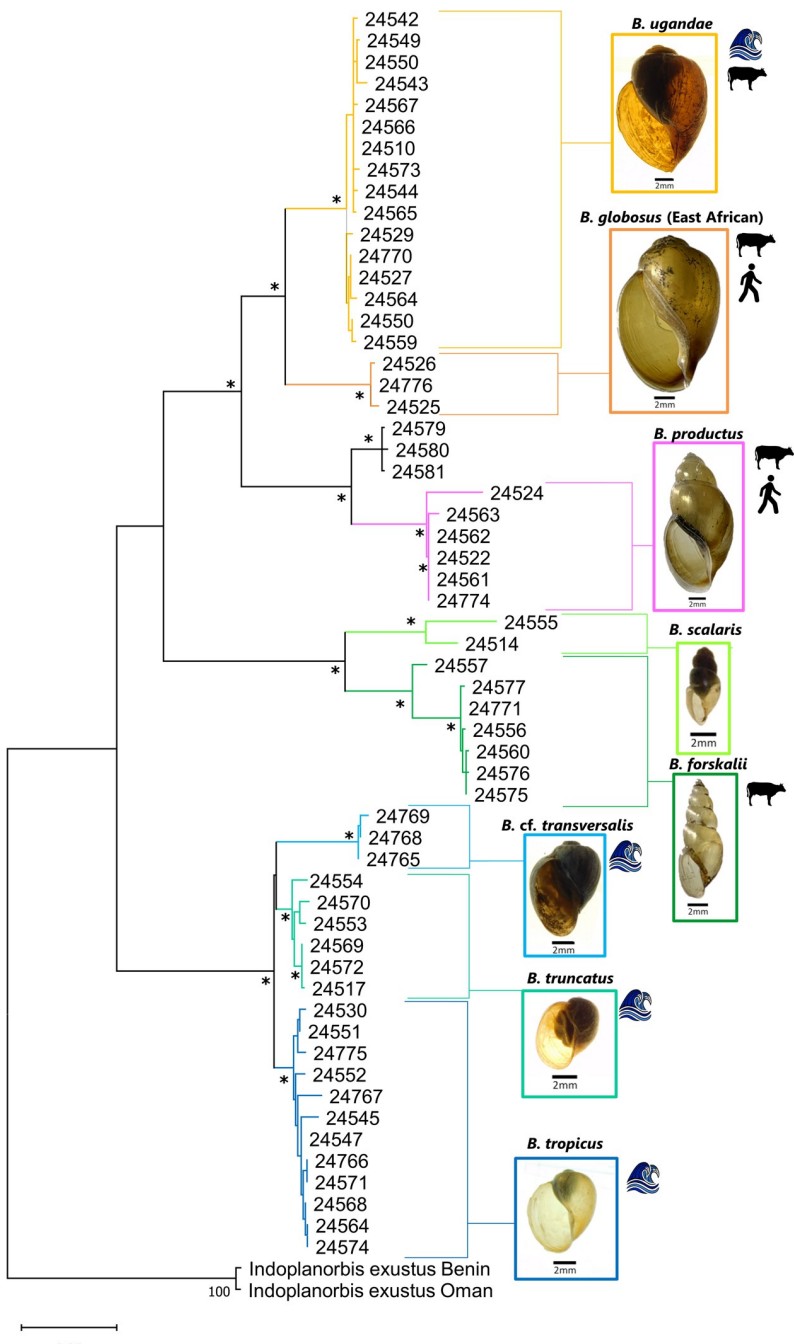

**Fig 2. Phylogenetic relationships of bulinids based on concatenated *cox1* + *16S* sequences.** Phylogenetic relationships of bulinids from this study and from GenBank based on 1163 bp of combined *cox1* and *16S* sequences inferred from ML analysis under the GTR+G+I model. Bootstrap values above 95 are indicated by an asterisk. Sequences generated during this study are listed by MSB:Host number. Additional information for specimens can be found in Table 1. Specimens recovered within the LVB by this study are color coded by species. *B. africanus* group species are in warm colors, *B. forskalii* group species in greens, and *B. truncatus/tropicus* group species in blues. Wave icons indicates species found within Lake Victoria. Cow icons indicate species with naturally occurring *S. bovis* infections. Human icons indicate species with naturally occurring *S. haematobium* infections.

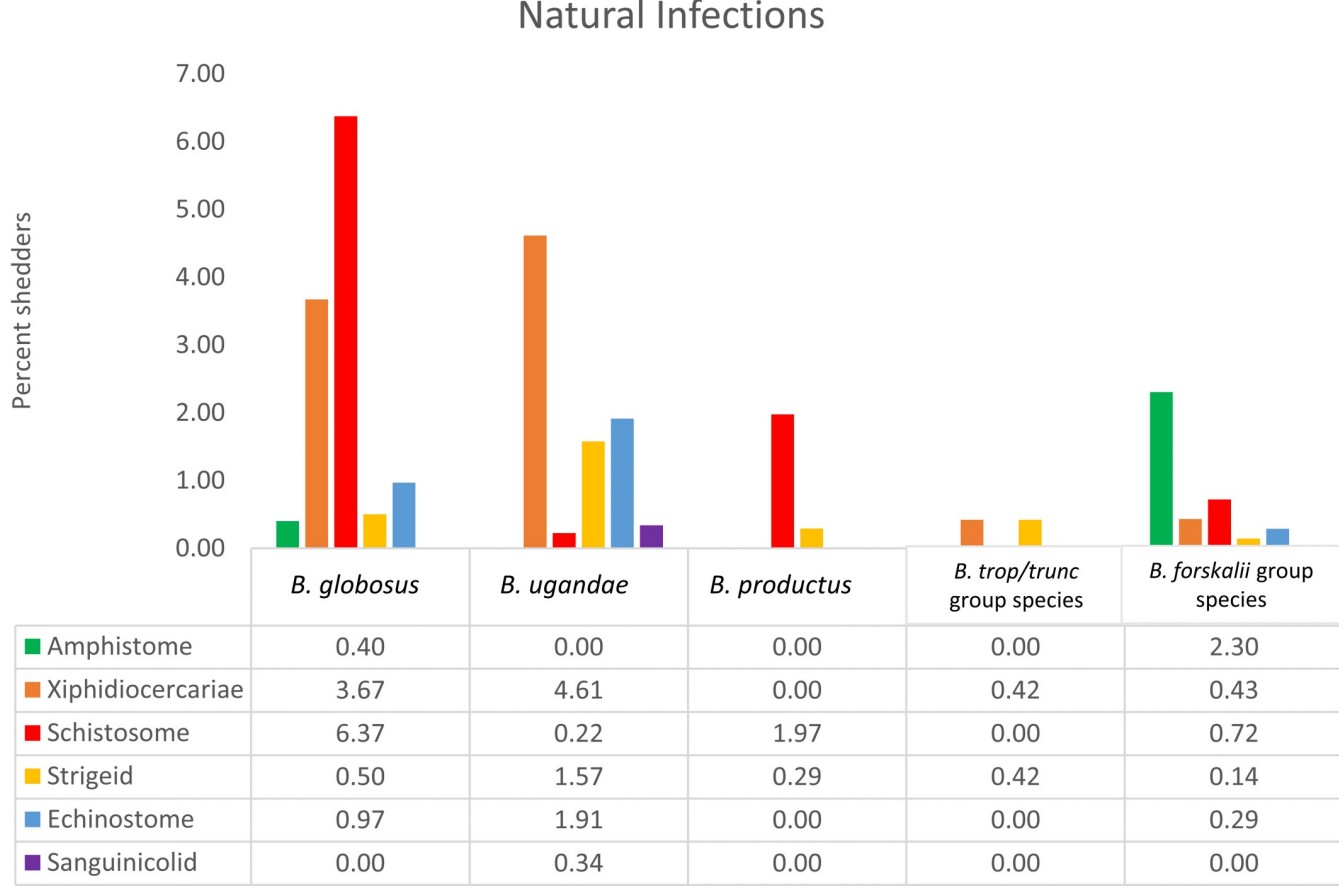

**Fig 3. Prevalence of natural infections among field collected bulinids.** Natural infection prevalence of 6 cercarial types shed from *Bulinus* snails collected between January 2014 and March 2021 from various localities in Kenya (S1 Table).

taxa coming from certain bulinid species including, for example, *B. globosus* and *B. ugandae* which are each known to host at least 2 species of echinostomes [57].

## Phylogenetic analysis of schistosomes

Partial *cox1* sequences provided by this study were primarily used to identify cercariae samples to the species level. Specimens were identified as either *S. haematobium* or *S. bovis*. *Cox1* and *ITS* sequences were used to examine cercariae samples for nuclear/mitochondrial discordance, which was not observed. Concatenated (*cox1 + ITS*) alignments were used to infer relationships among specimens (Fig 4), and information relating to specimens provided by this study can be found in Table 3. *S. haematobium* cercariae were recovered from *B. globosus* and *B. productus*; *S. bovis* cercariae were recovered from *B. globosus*, *B. productus*, *B. ugandae*, and *B. forskalii*. The phylogenetic analysis did not indicate affiliations between intermediate host species and schistosome genotypes for *S. bovis* specimens (Fig 4).

## Discussion

Our long-term goal is to understand the underlying biological processes that influence the complex interrelationships between bulinid snails and trematodes, especially schistosomes, in the LVB. Towards that end, we identified 8 distinct *Bulinus* taxa, 2 of which were naturally

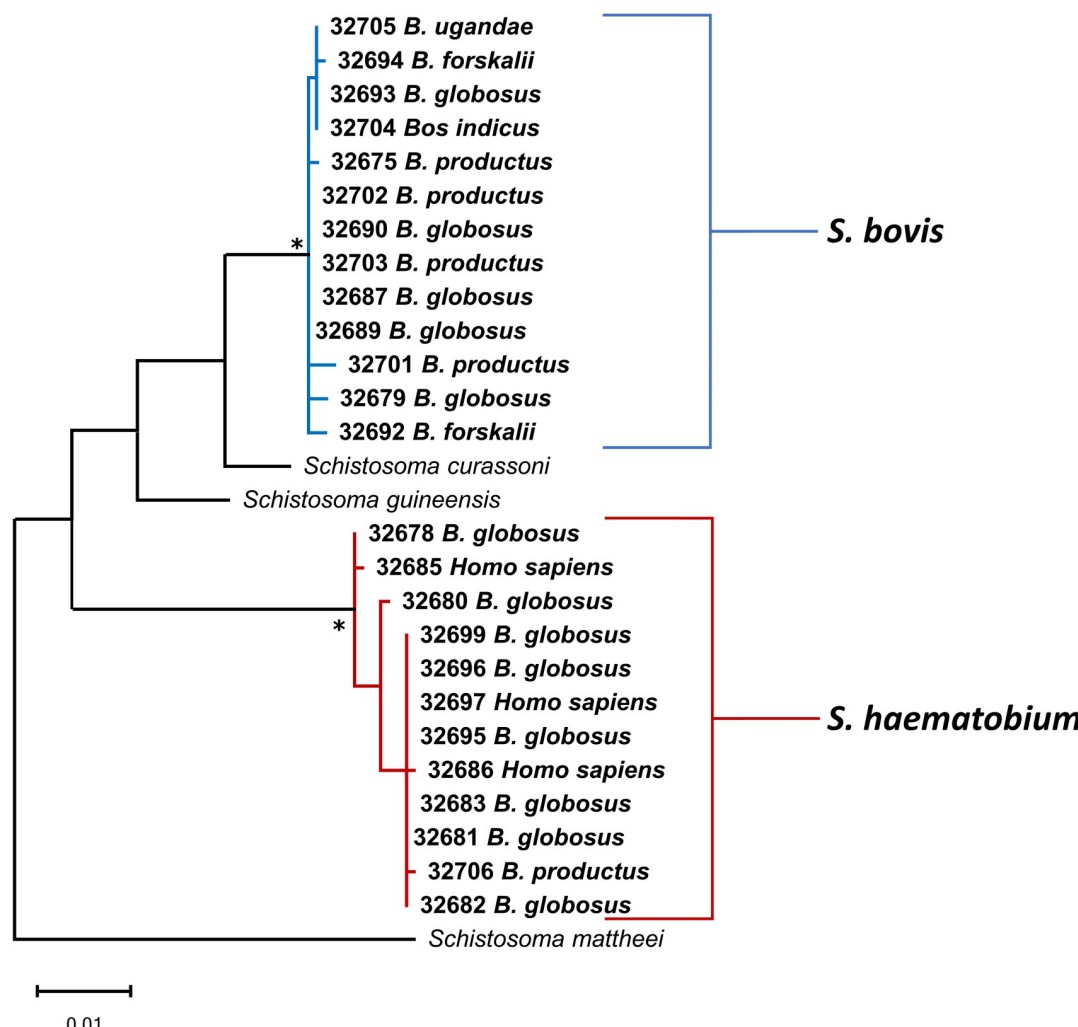

**Fig 4. Phylogenetic relationships among Kenyan schistosomes based on concatenated *cox1* + *ITS* sequences.** Phylogenetic relationships of schistosomes from this study and from GenBank based on 1166 bp of concatenated *cox1* + partial ITS1 + 5.8S + partial ITS2 sequences inferred from ML analysis. Bootstrap values over 95% are indicated by an asterisk. Specimens are listed by MSB:Para: number followed by the host species. Bolded sequences were generated during this study and additional information for specimens can be found in Table 3.

infected with *S. haematobium* (*B. globosus* and *B. productus*) and 4 of which were naturally infected with *S. bovis* (*B. globosus*, *B. productus*, *B. ugandae*, and *B. forskalii*). Additionally, 5 broad categories of non-schistosome cercariae were found among the bulinids collected.

With respect to the *B. africanus* group, the species reported thus far in the LVB are *B. africanus*, *B. globosus*, *B. nasutus*, *B. productus* (often referred to as *B. nasutus productus*), and *B. ugandae* [32,33]. Our results are similar but noteworthy in that we did not find *B. africanus* or *B. nasutus* (Figs 1 and 2). The presence of these taxa in areas we did not sample surely cannot be ruled out. We note that many of the previous identifications of *B. africanus* in the LVB were based largely on morphological criteria [32,60,84–86], which for reasons noted [36,87], may be particularly unreliable in East Africa.

Based on several lines of evidence, we suggest that the widespread occurrence of *B. africanus* in the LVB needs to be reconsidered. Chibwana *et al.* [50] did not report *B. africanus* from the LVB, and Pennance [35] noted the taxon he represented as *B. africanus* sp. 1 (representing locations in the LVB) is likely to be *B. ugandae*, with which we agree (see more below).

The phylogenetic inferences we generated for west Kenyan specimens of the related species *B. globosus* grouped with East African specimens designated as *B. globosus* on GenBank. Similar to observations of Kane *et al.* [38], we found (Fig 1) that type locality specimens for *B. africanus* [88] from Port Durban, South Africa, and for *B. globosus* [89] from Angola, belong to lineages separate from the East African *B. globosus* lineage. This suggests that the East African *B. globosus* is not conspecific with those from the type localities. In our phylogenetic analysis, representatives from the LVB did not group with the type-locality specimens for either *B. globosus* or *B. africanus*. Thus, future work should be done to characterize the *B. globosus* from LVB as it may be a different species. For convenience, we refer to it in this paper as *Bulinus globosus*. This line of thinking is consistent with other recent papers [46,50,75] that refer to a "*B. globosus* complex*", with multiple lineages represented in Kenya alone [46], and rendered even more complex than what we found when specimens from other parts of Africa are included [35,38].

*B. nasutus* has traditionally been divided into two subspecies: *B. nasutus nasutus* distributed mainly along the coastal provinces of Kenya and Tanzania, and *B. nasutus productus* distributed further inland from Uganda to Tanzania [36]. Morphometric analysis [90], enzyme analysis [33], and sequence analysis [38] suggested that the *B. nasutus* complex consists of two separate species, *B. nasutus* and *B. productus*. The *cox1 p*-distances we calculated (Table 2) for *B. nasutus* and *B. productus* were comparable to the *p*-distances found between other *Bulinus* species and justifies the separation of the *B. nasutus* complex into two species, which is also supported by a mitogenome analysis [35]. Enzyme analysis indicated both taxa were present in the LVB [33], but we found *B. nasutus* specimens only from central Kenya and *B. productus* only from the LVB. Additionally, *B. productus* was found exclusively in ephemeral pools and dams within the LVB while the related *B. globosus* was found primarily in streams (Table 1).

A more detailed understanding of the underlying systematics for *B. globosus* and *B. productus* is important because both are vectors of *S. haematobium* and have been implicated in natural infections by this study and by others [59,60,91–93] in the LVB. These taxonomic difficulties are especially unfortunate for the *B. globosus* species complex, members of which are important hosts for *S. haematobium* across tropical Africa; it remains awkward as to how to accurately name these important vector snails.

As noted above, another enigmatic member of the *B. africanus* species group is *B. ugandae*, widely reported throughout the LVB [84,85,94,95] but more rarely identified using molecular criteria. However, Chibwana *et al.* [50] identified a *Bulinus* sp. 2 which they suggested was *B. ugandae* based on analysis of their sequence data. Likewise, Pennance [35] identified *B. africanus* sp. 1, also suggesting it might be *B. ugandae*. More recently Zhang *et al.* [46] assembled the mitogenome of a *B. ugandae* sample from Lake Victoria. Based on examination of the shell photographs, similarities in habitat types, and phylogenetic analysis of sequences, we agree that these sequences represent *B. ugandae* specimens.

The phylogenetic relationships inferred in our study indicate that *B. ugandae* is sister to the East African *B. globosus* lineage as we have described it above. Figs 1 and 2 differ slightly in their topology, which may be resolved in the future with increased taxon sampling. However, both phylogenetic analyses support *B. ugandae* and East African *B. globosus* as separate lineages. In agreement with earlier studies [32,36,84,85], we did not find *B. globosus* in lacustrine habitats, while *B. ugandae* was found commonly from the shore of Lake Victoria or in marshes and swamps along the lakes edge.

It is of more than passing interest to correctly discriminate *B. ugandae* from *B. globosus* [96], and the application of molecular criteria is recommended. *Bulinus ugandae* is the only member of the *B. africanus* group not implicated in the transmission of *S. haematobium* [59,60]. The relationship between *B. ugandae* and *S. bovis* is more nuanced with field studies

suggesting that Kenyan *B. ugandae* is refractory to *S. bovis* [84] whereas other studies suggested that *B. ugandae* from Sudan or Uganda are vectors of *S. bovis* [94,97]. *Bulinus ugandae* from Western Kenya was found to be compatible with *S. bovis* in experimental infections [98], and we found *B. ugandae* to be naturally infected with *S. bovis* at two of our lakeshore study sites (S1 Table). The low prevalence of *S. bovis* in *B. ugandae* we observed may explain why some studies did not report natural infections. Alternatively, perhaps *S. bovis* relies on facilitation by other trematodes to successfully infect *B. ugandae* as has been reported in other bulinid species [10]. Pennance [35] also noted a natural infection of *B. ugandae* with an oft-overlooked member of the *S. haematobium* group, *S. kisumuensis*, previously known only from West Kenya based on anatomical characteristics and sequence data for adult worms recovered from rodents [24].

*B. ugandae* hosts a variety of other trematode species in the LVB (S1 Table and Fig 3). Amphistomes were not recovered during this study nor from a Tanzanian survey [85]. However, in Sudan, *B. ugandae* was found shedding amphistome cercariae [99], raising the possibility that significant intraspecific differences within *B. ugandae* may occur with resultant differences in compatibility with trematodes, further contributing to the complex patchwork of *Bulinus*-trematode compatibility so often noted.

*B. forskalii* species have received less attention than other bulinids in East Africa, likely because they are not associated with *S. haematobium* transmission in that area, unlike in West Africa [32,100,101]. Three *B. forskalii* group species: *B. forskalii*, *B. scalaris*, and *B. browni* have been reported from the LVB, and all have been observed to occur in sympatry [32]. In addition to finding *B. forskalii* commonly among our Kenyan samples, we found a juvenile of a second genetically distinct taxon that differed substantially from *B. forskalii*. It differed to a lesser extent from *B. scalaris* obtained from Ukerewe Island, Tanzania, the latter snail conforming conchologically to *B. scalaris* based on having rounded shoulders on the shell whorls [32]. The unknown juvenile tended to group with *B. scalaris* phylogenetically, yet intraspecific *p*-distances of these two sequences were higher than what has been reported within most *Bulinus* species (Tables 2 and S2). One possibility is that this snail is of the poorly known species *B. browni*, reported as being morphologically indistinguishable from *B. forskalii* but with unique enzyme banding patterns [102]. Its status remains uncertain as it has not been identified in any previous sequence-based analyses.

Neither *B. forskalii* nor *B. scalaris* are experimentally compatible with *S. haematobium* nor have been found to host natural infections in Western Kenya [59–61]. It is believed that *B. browni* similarly is not involved in transmission of *S. haematobium* [103], but both *B. forskalii* and *B. browni* have been implicated in the transmission of *S. bovis* [84,103,104]. These observations were supported by our findings which genetically identified *S. bovis* from natural infections in *B. forskalii*, yet we found no *S. haematobium* infections from any *B. forskalii* group snails. *B. forskalii* is known to vector a wide variety of other trematodes including amphistomes [55,105], echinostomes [106], and others [85]. Interestingly, the long periods of estivation that this species undergoes, which are associated with the ephemeral nature of their habitats, do not preclude it from frequently being parasitized by larval trematodes.

We found members of the *B. truncatus/tropicus* group only in Lake Victoria, an environment for which our accumulated taxonomic understanding for this species group is complicated. Based on morphological, enzymatic, and ploidy criteria, Brown [32] listed four members of the *B. truncatus/tropicus* group in Lake Victoria: *B. truncatus*, *B. tropicus*, *B. transversalis*, and *B. trigonus*. Chibwana *et al.* [50] recovered three taxa: *B. truncatus*, *B. tropicus* and *Bulinus* sp. 1 (considered to possibly be *B. trigonus* by the authors). Our efforts recovered three taxa: *B. truncatus*, *B. tropicus* and a third distinct taxon based on sequence criteria from *Bulinus* sp. 1 of Chibwana *et al.* [50]. Our third taxon most closely resembled *B. transversalis* conchologically [32,36], another bulinid species that remains poorly known.

The *cox1 p*-distances between *B. truncatus* and *B. tropicus* was the lowest among any two bulinid species we examined (Tables 2 and S2). Our presumptive *B. transversalis* and the presumptive *B. trigonus* of Chibwana *et al.* [50] differ to a greater extent from either *B. tropicus* or *B. truncatus*, and from each other, suggesting they are distinct species. The low *p*-distances between *B. truncatus* and *B. tropicus* has also been noted by others [39,43,107] and is somewhat paradoxical when considering their differences in ploidy, morphology and role as vectors of schistosomes.

Among the 245 individuals of the *B. truncatus/tropicus* group we examined, only 2 were positive for natural trematode infections (S1 Table and Fig 3). Neither Kenyan *B. truncatus* nor *B. tropicus* are known to vector local *S. haematobium* isolates [59,60,108]. However, Kenyan *B. truncatus* has been found compatible with allopatric *S. haematobium* isolates [60,109]. Experimental infections with what was likely a laboratory population of *B. transversalis* also proved refractory to East African *S. haematobium* infection [59]. *B. truncatus* has been found compatible with local isolates of *S. bovis* [84,110]. *B. tropicus* was found compatible with *S. bovis* only if it is previously infected with *Calicophoron microbothrium* [10,111]. No natural schistosome infections were documented for any member of the *B. truncatus/tropicus* group as part of our study.

As recently noted by Chibwana *et al.* [50], a range of *Bulinus* species are present in Lake Victoria and surrounding waters and they also noted that bulinid presence in the lake potentially implies the presence of *S. haematobium* and health risks from urogenital schistosomiasis for people living along the shore, or on the lake's islands. A considerable body of work has been undertaken over the years to examine the role of lake-associated bulinids in schistosome transmission (see the several papers cited above). Evidence from surveys and experimental infections, in agreement with data provided by this study, indicate that common lake species like *B. ugandae*, *B. tropicus* and *B. truncatus* are not found to be infected with local *S. haematobium* isolates, nor are members of the *B. forskalii* species group. Common *africanus* group species members like *B. productus* and *B. globosus* found in habitats other than lake shore are found to naturally host *S. haematobium*. The lake-dwelling *B. ugandae*, along with *B. forskalii*, *B. globosus* and *B. productus* have been found to naturally host *S. bovis* infections. At this time, unlike the situation for *S. mansoni*, the shorelines of Lake Victoria do not seem to pose a strong risk of *S. haematobium* infection.

As has been noted [60,61], East African *B. truncatus* are susceptible to what was historically described as *B. truncatus*-adapted isolates of *S. haematobium* common to Western Africa and Egypt, and introduction of isolates from these regions into the lake region might pose a new lake-borne *S. haematobium* problem. Likewise, introductions of exotic species into the lake, altered thermal or water quality regimes or changing populations of snail predators might change the current picture of *Bulinus* species representation in the lake, as they have in other African lakes [112].

Of further interest to us is to understand the puzzling underlying factors that dictate compatibility with *S. haematobium* of one *Bulinus* species, like *B. globosus*, whereas its close relative, *B. ugandae*, is seemingly refractory? This characteristic has a great deal to do with keeping *S. haematobium* transmission from occurring in the lake, thereby averting what could be a massive public health problem. Can this natural resistance to *S. haematobium* infection, if explained, in some way be used to lessen the vector potential of other bulinid species as a novel means of schistosomiasis control?

Similarly, we are interested in the characteristics of the west Kenyan *S. haematobium* isolates which favor or disfavor compatibility with certain bulinid species. *S. haematobium* isolates from across Africa have recently been shown to be genetically homogenous as compared to *S. bovis* [30], a characteristic that belies the evident heterogeneity in compatibility shown by

*S. haematobium* across Africa with respect to *Bulinus* species use. One possible explanation is that all *S. haematobium* isolates tested, with the exception of the Madagascar isolate, have been found to contain varying levels of *S. bovis* introgression in their genomes [28–30]. It will be of interest to determine if the content of such introgressed regions influence the compatibility of *S. haematobium* to different *Bulinus* species.

Other avenues of interest for disentangling the *Bulinus*-schistosome compatibility include the role of symbionts, such as annelids (*Chaetogaster*), which may prey upon the miracidia or cercariae of trematodes, thereby reducing transmission [9]. Chaetogasters are particularly conspicuous on field-derived specimens of *Bulinus* [113] and deserve further scrutiny with respect to their impact on influencing infection success of schistosome miracidia.

We are similarly interested in applying the notion of coevolutionary hot and cold spots [114,115] to Lake Victoria shorelines, owing to their intense use by many host species potentially carrying many trematode species [11]. Shoreline locations have been considered coevolutionary hot spots and may dictate certain type of immune or other avoidance strategies by snails to avoid high infection rates. In contrast, deep water locations are considered coevolutionary cold spots because fewer host species (and attendant trematodes) frequent them, which might select for different response strategies among snails living there. We are similarly interested to learn if species like *B. forskalii* that so often are found in ephemeral habitats and known to be preferential self-crossers [116] have fundamentally different strategies for dealing with pathogens like trematodes than snails that occupy far more stable conditions, like the shoreline habitats of Lake Victoria.

## Conclusions

Based on *cox1* sequence data, we found 8 distinct taxa of *Bulinus* in our west Kenyan sampling locations: *B. globosus*, *B. productus*, *B. ugandae*, *B. forskalii*, presumptive *B. scalaris*; *B. tropicus*, *B. truncatus* and presumptive *B. transversalis*. We found natural infections of *S. haematobium* in *B. globosus* and *B. productus*, and the ruminant schistosome *S. bovis* in these two species as well as in *B. ugandae* and *B. forskalii*, confirming the vector role for these species outlined in previous studies. We highlight the importance of providing molecularly-based identification, particularly in regards to discriminating *S. haematobium* vector species like *B. globosus* from related non-vector species like *B. ugandae*. Several outstanding issues with respect to *Bulinus* systematics were noted: the lack of bona fide *B. africanus* in our samples and the presence of a "*B. globosus* complex" requiring further resolution; the status of *B. productus* as a distinct species from *B. nasutus*; and the need for further collection and resolution among species in both the *B. forskalii* and *B. tropics/truncatus* groups, the latter especially as it pertains to the LVB. The complex patterns of *Bulinus-Schistosoma* compatibilities noted argue for more in-depth study to understand factors dictating the underlying patterns that, at least thus far, have fortuitously kept the immediate shoreline and waters of Lake Victoria largely free of *S. haematobium* transmission.

## Supporting information

**S1 Fig. Collection Locations within the Lake Victoria Basin, Western Kenya.** ExpertGPS Basemap of collection locations within the Lake Victoria Basin in Western Kenya for bulinid snails. Information regarding samples from these locations can be found in Tables 1 and S1. Base map and data from OpenStreetMap and OpenStreetMap Foundation. Base-layer retrieved from https://www.openstreetmap.org/relation/192798.
(TIF)

**S1 Table. Natural Infections in Bulinids.** The number of snail specimens examined and number of observed trematode infections is listed per species and per survey sites. Total number of snail specimens and percent of individuals infected (in parentheses) is listed per cercarial type. Map of collection locations can be found in S1 Fig. Habitat types are: LS = lakeshore, L = lake, R = river, EP = ephemeral pond, D = Dam, S = Swamp.
(XLSX)

**S2 Table. Intra- and Interspecies *p*-distance values of concatenated partial *cox1 + 16S* of 58 bulinid sequences.** Bolded values are intraspecies *p*-distance values.
(XLSX)

## Acknowledgments

We thank Ibrahim Mwangi, Joseph Kinuthia, Geoffrey Maina, and Boaz Oduor for their assistance with the collection of field samples and snail maintenance. We also thank Dr Stephen Munga, the Deputy Director of the Center for Global Health Research, Kenya Medical Research Institute (KEMRI), for providing the laboratory space to conduct these experiments. Technical assistance at the University of New Mexico Molecular Biology Facility was supported by the National Institute of General Medical Sciences of the National Institutes of Health under Award Number P30GM110907. This work was published with the approval of the Director-General, KEMRI.

The content for this paper is solely the responsibility of the authors and does not necessarily represent the official views of the National Institutes of Health.

## Author Contributions

**Conceptualization:** Caitlin R. Babbitt, Eric S. Loker.

**Data curation:** Caitlin R. Babbitt.

**Formal analysis:** Caitlin R. Babbitt.

**Funding acquisition:** Eric S. Loker.

**Investigation:** Caitlin R. Babbitt, Martina R. Laidemitt, Martin W. Mutuku, Polycup O. Oraro.

**Methodology:** Caitlin R. Babbitt, Martina R. Laidemitt.

**Project administration:** Gerald M. Mkoji, Eric S. Loker.

**Resources:** Gerald M. Mkoji, Eric S. Loker.

**Supervision:** Gerald M. Mkoji.

**Validation:** Sara V. Brant.

**Writing – original draft:** Caitlin R. Babbitt, Eric S. Loker.

**Writing – review & editing:** Caitlin R. Babbitt, Martina R. Laidemitt, Sara V. Brant, Eric S. Loker.

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
