## [Decision Letter · Decision Letter 0]

14 Nov 2022

Dear Ms. Babbitt,

Thank you very much for submitting your manuscript "Bulinus snails in the Lake Victoria Basin in Kenya: systematics and their role as hosts for schistosomes" for consideration at PLOS Neglected Tropical Diseases. As with all papers reviewed by the journal, your manuscript was reviewed by members of the editorial board and by an independent reviewer. The reviewer appreciated the attention to an important topic. Based on the review, we are likely to accept this manuscript for publication, providing that you modify the manuscript according to the review recommendations. 

Sincerely,

Brianna R Beechler, Ph.D., DVM

Academic Editor

Esther Schnettler

Section Editor

The reviewer recommends some minor changes that will improve the manuscript.

Reviewer's Responses to Questions

**Key Review Criteria Required for Acceptance?**

**Methods**

-Are the objectives of the study clearly articulated with a clear testable hypothesis stated?

-Is the study design appropriate to address the stated objectives?

-Is the population clearly described and appropriate for the hypothesis being tested?

-Is the sample size sufficient to ensure adequate power to address the hypothesis being tested?

-Were correct statistical analysis used to support conclusions?

-Are there concerns about ethical or regulatory requirements being met?

Reviewer #1: (No Response)

**Results**

-Does the analysis presented match the analysis plan?

-Are the results clearly and completely presented?

-Are the figures (Tables, Images) of sufficient quality for clarity?

Reviewer #1: (No Response)

**Conclusions**

-Are the conclusions supported by the data presented?

-Are the limitations of analysis clearly described?

-Do the authors discuss how these data can be helpful to advance our understanding of the topic under study?

-Is public health relevance addressed?

Reviewer #1: (No Response)

**Editorial and Data Presentation Modifications?**

Reviewer #1: (No Response)

**Summary and General Comments**

Reviewer #1: The manuscript submitted to PLOS NTDs by Babbitt et al reports on the presence and great diversity of Bulinus spp. snails in the Lake Victoria basin region in Kenya. This is of particular interest, since as highlighted in the far- and few-between studies looking at these snails, we know a great diversity exists, but little has been done on a molecular level to unravel the diversity of these species, and especially in the context of schistosome transmission. Specifically, on schistosome transmission, this study reports from a large sample size of snails, that transmission of urogenital schistosomiasis is likely not occurring within the waters of the lake itself, providing some reassurance of previous findings. 

This is a great study adding a lot of important molecular information for the LVB and lacustrine Bulinus species and the schistosome species transmitted in this region. I highly recommend the paper is published in PLOS NTDs after the following minor recommendations are made or appropriately responded to.

General comments

Final sentence of the abstract could be a little unclear for some readers. Define what pattern is referred to (species distribution? Schistosome compatibility? Lake vs non-lake?) and also what biological mechanisms are meant (i.e. are you referring to genes involved in resistance to schistosomes? Those that help with species ecology and for targeted snail control?). I fully appreciate that the authors want to be broad with this closing statement, but to me it does not provide enough information to make worthwhile including currently and I believe can be reworked to something much more impactful. 

I really enjoyed reading both the introduction and the rich discussion, full of good insights and relating back to previous studies to summarize where we are at with the Bulinus genus in East Africa.

S1 Table and Table 1 – I find the way information is split across Table 1 and S1 Table a little bit frustrating. To make it easier for the reader, I believe an S1 table that has a summary of each site (i.e. rows), its coordinates, the waterbody type and number of Bulinus spp. (including break down of species) and number infected (and %) the number molecularly identified, summarized together would be more informative. Summarizing the data by site (i.e. rows), and breaking down the number of snails, and infected snails, collected within each species as columns would for me provide an easier reference than how laid out currently. Table 1 could then remain as is but habitat type, longitude and latitude could be removed as would be contained in S1 table, or another simplified summary table..

As an addition, it would also be good to have the temporal breakdown of when the snails were collected, i.e. so that in future it can be used to identify snail abundance changes across seasons / years, as we know that this can be really important with the changing LVB. Although some of this temporal information is given in Table 1, it is not clear if this represents all the snails from collections from those sites or not. As all this information is there in the manuscript as is, this comment is more of a suggestion than a necessary change.

A map Figure would be a wonderful addition to this manuscript to help represent where collections were made, the distance between sites, and where species were found to related to phylogenetic analysis. Would be great in the manuscript or even as a supplementary figure.

Line by line comments

52 – Maybe a brief mention of S. mansoni group and Biomphalaria spp. presence?

59 – ‘DNA sequence based’

59 – ‘shed from infected snails’ - to clarify that looking at patent infections only. 

97 – ‘includes 9 species: ..’

130 – change in reference style?

131 – think this paragraph should be attached to previous one to lead on from the end of the last paragraph ‘i.e. Previously… More recently..’ – since not starting a ‘new’ point. 

131-137 – Not sure however if all this discussion necessary related to hybrids here. Suggest simplifying by removing last two sentences.

139 – From reading rest of paper – doesn’t seem that too much focus is given to ‘other’ trematodes outside schistosomes except for morphologically identifying to genus and brief part in results 353-358 – therefore I would reword this sentence to make clear this is really focusing on Bulinus and schistosomes, with some insight into trematodes too.

144 – for the readers ease, would be nice to have this list split up into the species groups they represent too?

165 – Sentence to use in reworking final abstract sentence? 

171 – Gives the impression that localities are specifically defined in S1 Table, yet they are not really, just names This is provided in the my general comment above, but you could provide long / lat here in S1 table. But another suggestion might be just to include a summary table by site listening the name, coordinates, water body type and number of Bulinus spp. collected? 

172 – Collections span from Jan 2014 to ?? Could specify here for this study at least, even if collections are continuing.

178 – Could it be more specifically stated how this combined 150m was achieved, or better, point to it in reference cited on line 173 i.e. Mutuku 2019 if it is contained in here

190 – Why the S. haematobium have collected from humans here is not clear. Can see from later in paper it is to compare with those shed from Bulinus in phylogenetic analysis. Worth mentioning that here in my opinion to be clear, as my thoughts were that experimental exposures may be taking place. Would be great to do challenges of the Lake Bulinus with schistosomes in the future.. Also could add, how many miracidia collected from x number of individuals?

213 – Could it be specified, maybe in table 1 – when alternative COR722b primers were used for amplification? Was this due to sequence diversity in particular species? Could be useful for reference in future studies. 

223 – were individual cercariae therefore removed from pooled ethanol preserved specimens? 

248 – references to associated studies could be included for the genbank accessioned used. 

261 – provide accession numbers for sequences in current study here too?

270 – Can you list the number of specimens identified to each species / species group in the main text. Also not clear to all readers in table S1 which parts represent species groups and which species – I presume B. forskalii listed in S1 Table is species group and not representing species alone, as must include the B. scalaris identified as noted in table 1 and later in manuscript? Denote that B. truncatus / tropicus group and forskalii group are therefore identified to species group level in text and in S1 table (unless I am misinterpreting?). 

272 – Highest / lowest S. haematobium prevalence observed from where? Of interest and could be mentioned in main text here briefly?

289 – Denote in table and legend which samples from archived specimens? ‘*’ i.e. B. nasutus? 

320 – As for Table 2 – could be good practice to include references to the reference sequences used in the phylogenetic tree? See earlier point in methods too.

338 – Reference for Indoplanorbis sequences in Figure?

348 – Still would be interested to see how schistosome infections vary over time or specific sites mentioned in text. Would help highlight details in Table S1 regarding sites with lots of infected snails. No temporal detail for snail collections of infections included currently, but I believe could easily be added.

433 – Last sentence here seems a bit of a stub – understand where going with this paragraph in saying one might consider these a very wide complex of species, but could this paragraph be reworked to make more clear?

454 – Bulinus in full at sentence start. 

545 – this attribute – can it be expanded on, hypothesized just in a few words? i.e. genetic resistance in snails or something else? Could this also be related to what is lead into the discussion in line 556 onwards?

PLOS authors have the option to publish the peer review history of their article (what does this mean?). If published, this will include your full peer review and any attached files.

Reviewer #1: Yes: Tom Pennance

Figure Files:

Data Requirements:

Reproducibility:

References

---

## [Decision Letter · Decision Letter 1]

20 Jan 2023

Dear Ms. Babbitt,

We are pleased to inform you that your manuscript 'Bulinus snails in the Lake Victoria Basin in Kenya: systematics and their role as hosts for schistosomes' has been provisionally accepted for publication in PLOS Neglected Tropical Diseases.

Best regards,

Brianna R Beechler, Ph.D., DVM

Academic Editor

Esther Schnettler

Section Editor

The authors have improved the manuscript in response to the previous review and it is now acceptable for publication.

Reviewer's Responses to Questions

**Key Review Criteria Required for Acceptance?**

**Methods**

-Are the objectives of the study clearly articulated with a clear testable hypothesis stated?

-Is the study design appropriate to address the stated objectives?

-Is the population clearly described and appropriate for the hypothesis being tested?

-Is the sample size sufficient to ensure adequate power to address the hypothesis being tested?

-Were correct statistical analysis used to support conclusions?

-Are there concerns about ethical or regulatory requirements being met?

Reviewer #1: (No Response)

**Results**

-Does the analysis presented match the analysis plan?

-Are the results clearly and completely presented?

-Are the figures (Tables, Images) of sufficient quality for clarity?

Reviewer #1: (No Response)

**Conclusions**

-Are the conclusions supported by the data presented?

-Are the limitations of analysis clearly described?

-Do the authors discuss how these data can be helpful to advance our understanding of the topic under study?

-Is public health relevance addressed?

Reviewer #1: (No Response)

**Editorial and Data Presentation Modifications?**

Reviewer #1: (No Response)

**Summary and General Comments**

Reviewer #1: I am happy that all the suggestions made for revision have been attended to and look forward to seeing the final manuscript published in PLOS NTDs.

PLOS authors have the option to publish the peer review history of their article (what does this mean?). If published, this will include your full peer review and any attached files.

Reviewer #1: **Yes: **Tom Pennance

---

## [Editor Report · Acceptance letter]

7 Feb 2023

Dear Ms. Babbitt,

We are delighted to inform you that your manuscript, "*Bulinus* snails in the Lake Victoria Basin in Kenya: systematics and their role as hosts for schistosomes," has been formally accepted for publication in PLOS Neglected Tropical Diseases.

Best regards,

Shaden Kamhawi

co-Editor-in-Chief

Paul Brindley

co-Editor-in-Chief
